# Effects of Red and Blue Laser Irradiation on the Growth and Development of *Ostrinia furnacalis*

**DOI:** 10.3390/insects16090906

**Published:** 2025-08-29

**Authors:** Xuemei Liang, Xintong Dai, Li Qin, Xiao Feng, Ge Chen, Minglai Yang

**Affiliations:** 1College of Information Technology, Jilin Agricultural University, Changchun 130118, China; xuemeil@jlau.edu.cn (X.L.); daixintong20@163.com (X.D.); 2Changchun Institute of Optics, Fine Mechanics and Physics, Chinese Academy of Sciences, Changchun 130033, China; qinl@ciomp.ac.cn; 3College of Plant Protection, Jilin Agricultural University, Changchun 130118, China; fengxiao@mails.jlau.edu.cn (X.F.); ge_chenge@163.com (G.C.)

**Keywords:** laser irradiation, *Ostrinia furnacalis*, photobiological regulation, developmental delay, physical control, red and blue lasers

## Abstract

The Asian corn borer (*Ostrinia furnacalis* (Guenée)) is a highly destructive pest affecting crops worldwide. Traditional reliance on chemical control causes significant pollution. This study explores a novel laser-based approach to suppress the pest. By rearing larvae under lights of varying intensity and spectrum, we found that blue laser light significantly prolonged larval development and reduced pupation rates. This delay in metamorphosis and decline in moth emergence imply severely weakened reproductive capacity. These findings highlight a promising eco-friendly alternative to pesticides. Laser lights could serve as an innovative tool for farmers to protect crops from the Asian corn borer, potentially reducing dependence on chemical methods. Further research may optimize this strategy for field applications.

## 1. Introduction

The corn borer includes two major species, the Asian corn borer (*Ostrinia furnacalis* (Guenée)) and the European corn borer (*Ostrinia nubilalis* (Hübner)), with the former causing more severe damage in China. The Asian corn borer (hereinafter referred to as the corn borer) belongs to the family *Crambidae* of the order Lepidoptera and is widely distributed across the country [1]. It is a destructive agricultural pest that feeds on at least 69 plant species, including maize, sorghum, and cotton [2]. Larvae initially feed on heart leaves and tassels of maize, but from the third instar onward, they bore into plant stems, leading to stem lodging, grain damage, and increased vulnerability to secondary infections such as ear rot [3,4]. These injuries significantly reduce yield and grain quality, posing a threat to maize production security.

Chemical pesticides remain the primary method for controlling corn borer populations. However, long-term use has resulted in decreased effectiveness and considerable ecological impact. Pesticide residues accumulate in soil, disrupting microbial communities and reducing soil fertility [5]. In addition, surface runoff and leaching can transport pesticides into aquatic systems, contaminating groundwater and surface water, threatening aquatic life, and compromising drinking water safety [6]. While biological control offers an eco-friendly alternative, its effectiveness is often constrained by environmental conditions and pest dynamics [7]. These limitations underscore the urgent need for innovative control strategies. Among them, physical control methods—particularly those based on light regulation—are gaining attention for their potential to offer safe, effective, and sustainable pest management solutions.

Laser technology, celebrated for its monochromaticity and high energy efficiency, has seen profound applications across various domains, including agriculture [8,9]. In agricultural breeding, specific wavelength laser irradiation is employed to induce mutagenesis in crops such as soybeans, wheat, and fruit trees, thereby enhancing genetic traits [10,11]. Plant physiological responses to light can be optimized through precise spectral simulations, which have been shown to increase yield by manipulating photoperiod responses [12] and to mitigate UV-B radiation-induced damage [13]. Rapid phenotypic detection systems based on spectral analysis provide early warning for pests and diseases. For instance, Sankaran et al. utilized laser-induced breakdown spectroscopy (LIBS) to analyze citrus leaves for anomalies, including diseases such as Huanglongbing (HLB) and nutrient deficiencies. The biothermal effects of intense lasers can disrupt protein structures, leading to the inactivation of viruses and bacteria, the stimulation of anti-inflammatory responses, and the control of weeds and pests [14]. Directional laser technology offers a non-chemical approach to pest management by selectively inhibiting weed growth [15]. Furthermore, lasers can activate graphite materials to enhance their adsorption capacity, addressing water pollution issues [16]. These advancements underscore the potential of laser technology to support sustainable agricultural development.

Light is a critical environmental factor that regulates insect growth and development. It influences multiple biological processes, including developmental duration, survival and diapause rates, reproductive behavior, and predator–prey interactions [17,18]. Studies have shown that different wavelengths and intensities of light can elicit species-specific physiological responses. For example, blue light has been reported to significantly disrupt egg hatching patterns in *Empoasca onukii*, resulting in reduced hatch rates [19]. Similarly, exposure to specific light environments has been shown to influence developmental timing and reproductive capacity in various insects [20,21,22].

Red and blue light, in particular, have received extensive attention. Red light is generally associated with inhibitory effects on insect development. For instance, Shimoda and Honda demonstrated that red light delayed larval development and decreased pupation rates in certain Lepidoptera [23]. On the other hand, blue light exhibits more complex effects. While it can attract insects and alter circadian rhythms [24], prolonged exposure may induce developmental abnormalities. Hori et al. (2014) observed that blue light suppressed reproductive behaviors in *Drosophila melanogaster* [25]. At a wavelength of 420 nm, blue light reduced survival, pupation, and emergence rates in *Mythimna separata* (walker) by 72.00%, 65.67%, and 72.87%, respectively, while extending larval and pupal stages by 12 and 11.4 days [26]. Additionally, Dong (2018) reported significant growth inhibition in wheat moth (*Sitotroga cerealella*) under blue light conditions [17]. In fall armyworm (*Spodoptera frugiperda*), mating rates were significantly reduced during specific dark periods when exposed to blue light [27]. Taken together, current evidence highlights the diverse effects of red and blue light on insect biology, which vary according to species, developmental stage, wavelength, and exposure conditions. These findings underscore the potential of light-based methods in pest management, while also indicating the need for further mechanistic studies to optimize practical applications.

Despite the extensive research on the effects of non-coherent light sources on insects, the application of coherent laser light in pest control remains underexplored. The unique properties of laser light, such as coherence, monochromaticity, and high energy density, suggest that it may have distinct biological effects compared to traditional light sources.

This study investigates the effects of red, blue, and combined laser light treatments on the developmental process of the corn borer (*O. furnacalis*). Through controlled experiments, the influence of these treatments on pupation duration, adult emergence rate, and oviposition behavior was systematically evaluated. The aim is to explore light-based pest control strategies as innovative, environmentally friendly alternatives to conventional chemical pesticides.

## 2. Materials and Methods

### 2.1. Insect Source and Rearing Conditions

The larvae of corn borer used in this experiment were originally sourced from Guangdong, China, and reared under controlled laboratory conditions (temperature: 25 ± 3 °C; relative humidity: 65 ± 5%; photoperiod: 12 h:12 h). Upon hatching, larvae were placed into round rearing containers (diameter: 10 cm; height: 5.5 cm). After one week, they were transferred to larger containers (diameter: 15 cm; height: 15 cm) for continued development. An artificial diet was used as the feeding medium, with its composition detailed in Table 1.

### 2.2. Lighting Equipment and Construction of Artificial Darkroom

In this experiment, a laser plant growth light was used, consisting of red laser light with a wavelength of 660 nm and blue laser light with a wavelength of 450 nm (15W, CXL1-X0220-0601, Zhejiang Changxin Optoelectronics Science and Technology Co., Hangzhou, China). Unlike traditional LED light sources, the laser light exhibits a higher photoelectric conversion efficiency (laser ≥ 40%, LED ≤ 25%) and reduces energy consumption per unit light intensity by approximately 50%. A detailed comparison of energy consumption is presented in Table 2 (data source: Zhejiang Changxin Optoelectronics Technology Co., Ltd.).

For this study, a custom-designed double-layer experimental darkroom was constructed, with each layer measuring 43 cm × 33 cm × 38 cm (see Figure 1 for schematic illustration). The exterior was covered with opaque black fabric to ensure complete light isolation. A laser light source was mounted at the top of each chamber using nylon cable ties, positioned 30 cm above the insects. This layered design enabled independent treatment conditions for each group, while maintaining precise light control and facilitating convenient experimental operation.

### 2.3. Laser Light Source Parameters

The selection of laser light intensities in this study was guided by empirical results from prior plant supplemental lighting experiments. Trials with rice seedlings and horticultural crops indicated that a minimum photosynthetic photon flux density (PPFD) of approximately 0.5 μmol/m^2^/s was required to elicit significant biological responses. In greenhouse applications, this threshold could only be achieved at the maximum output setting (level 9), which is why plant experiments often adopted red–blue ratios such as 9:6 or 9:3.

However, preliminary tests in the current insect study revealed that such high intensities caused acute larval mortality due to photothermal effects rather than specific developmental inhibition. To avoid this confounding outcome, moderate intensity levels were chosen for the formal experiments, namely red light at levels 4 and 6 and blue light at levels 2, 4, and 6. The combined red–blue groups were set accordingly. These treatment levels ensured that the light intensity was sufficient to induce photobiological responses while minimizing non-specific lethality.

Experimental subjects were therefore assigned to eight treatment groups with varying laser light conditions. The corresponding absolute irradiance values (PAR and PPFD) are detailed in Table 3. The treatment design was as follows:

1. Control Group: Larvae were reared under natural light conditions and served as the baseline for comparison.

2. Red Light Treatment Groups:

(1) R4: Exposed to red laser light at intensity level 4 (PPFD Red = 13.851 μmol/m^2^/s; see Table 3).

(2) R6: Exposed to red laser light at intensity level 6 (PPFD Red = 25.138 μmol/m^2^/s; see Table 3).

3. Combined Red and Blue Light Treatment Groups:

(1) RB2: Exposed to combined red and blue laser light with blue light intensity set at level 2 (PPFD Red = 9.489 μmol/m^2^/s; PPFD Blue = 9.489 μmol/m^2^/s; see Table 3).

(2) RB4: Exposed to combined red and blue laser light with blue light intensity set at level 4 (PPFD Red = 16.813 μmol/m^2^/s; PPFD Blue = 7.303 μmol/m^2^/s; see Table 3).

4. Blue Light Treatment Groups:

(1) B2: Exposed to blue laser light at intensity level 2 (PPFD Blue = 3.686 μmol/m^2^/s; see Table 3).

(2) B4: Exposed to blue laser light at intensity level 4 (PPFD Blue = 12.114 μmol/m^2^/s; see Table 3).

(3) B6: Exposed to blue laser light at intensity level 6 (PPFD Blue = 16.230 μmol/m^2^/s; see Table 3).

### 2.4. Developmental Duration, Reproductive Period, and Reproductive Capacity

To investigate the pupation and oviposition behavior of the corn borer under different laser irradiation conditions, larvae were assigned to eight treatment groups. Each group consisted of 100 larvae, divided into two containers (50 larvae per container), and each treatment was replicated three times.

To evaluate the effects of red and blue laser treatments on development and reproduction, data were continuously recorded throughout the insect’s major life stages. Egg hatching was monitored four times daily (08:30, 11:00, 14:30, and 17:00), starting immediately after egg papers were replaced, sealed in polyethylene bags, and placed in a darkroom. Observations continued until larval emergence. Hatching time was calculated using the following formula:Hatching time (days) = Larval hatching time − Egg paper replacement time(1)

Larval development was recorded twice daily (09:00 and 15:00) from hatching until pupation. The duration of the larval stage was calculated as follows:Larval stage duration (days) = Pupation date − Hatching date(2)

Each pupa was individually transferred to a rearing box containing a moistened cotton pad to prevent cannibalism and was maintained under the same light conditions as during the larval stage. The number of new pupae was recorded daily until no further pupation occurred. The pupation rate was determined as follows:Pupation rate (%) = (Number of pupated larvae / Initial number of larvae) × 100%(3)

To assess reproductive performance, unmated adults within 24 h of emergence were paired (one female and one male per pair; up to five pairs per treatment) and kept in darkrooms under their respective light conditions. Egg papers (15 × 15 cm) were replaced daily, sealed in plastic bags, and stored at 70 ± 5% relative humidity. Daily egg production was recorded, and egg bags were returned to the same environment for hatching observation. After all females had died, the total number of eggs laid and the oviposition period were determined. Hatching rate was calculated based on larval emergence using the following formula:Hatching rate (%) = (Number of hatched larvae/Total number of eggs) × 100%(4)

### 2.5. Data Analysis and Processing

The experimental data were analyzed using IBM SPSS Statistics v27 https://www.ibm.com/cn-zh/products/spss-statistics/campus-editions (accessed on 25 August 2025) for statistical analyses and Origin v2024 https://www.originlab.com/demodownload.aspx (accessed on 25 August 2025) for graphing. A one-way analysis of variance (ANOVA) was performed to compare the effects of different light conditions on various parameters, including hatching time, larval stage duration, hatching rate, pupation rate, and egg production. Statistical significance was set at *p* < 0.05. If the ANOVA revealed significant differences (*p* < 0.05) among groups, Tukey’s Honest Significant Difference (HSD) post-hoc test was applied to identify which specific groups differed from each other. The results were presented with letter groupings (a, b, c, etc.) to indicate significant differences between groups, where different letters represent significantly different groups, and identical letters indicate no significant differences.

## 3. Results

### 3.1. Effects of Red and Blue Lasers on the Hatching Time of the Corn Borer Eggs

The hatching times of insect eggs under different lighting conditions are shown in Figure 2. The laser treatment group showed significant differences compared to the CK group. Among them, the red light treatment group R4 had a 1-day delay in egg hatching, and a small portion of the eggs in R4 did not hatch; the eggs in R6 had a 0.8-day delay. In the blue laser treatment group, the hatching times of B2 and B4 eggs were delayed by 0.6 days compared to those under normal light, while B5 eggs were delayed by 0.5 days. In the red and blue laser combination treatment group, RB2 eggs experienced a 0.6-day delay compared to the CK group, and RB4 eggs were delayed by 0.9 days. Figure 3 shows images of the insect eggs from each treatment group at the time of larval emergence in the CK group.

### 3.2. Effects of Red and Blue Lasers on the Larval Stage of the Corn Borer

The larval stages of corn borers under different lighting conditions are shown in Figure 4. The laser treatment group showed significant differences compared to the CK group (*p* < 0.001). The larval durations for R4 and R6 treatments were approximately the same as those under normal light conditions, with no significant differences observed (*p* > 0.05). However, the number of pupating larvae in B2, B4, B6, RB2, and RB4 treatments significantly differed from those in the CK group (*p* < 0.001). Notably, in the B4 treatment, some larvae failed to pupate even after 40 days post-hatching, and in the treatments B2, B6, RB2, and RB4, there were larvae that did not pupate for over 30 days. Figure 5 displays images of corn borer larvae under blue light treatments that exceeded 30 days in their larval stage.

### 3.3. Effects of Red and Blue Lasers on the Pupation of the Asian Corn Borer

The numbers of pupae under different lighting conditions are shown in Figure 6. The pupation numbers for R4 and R6 treatments were approximately the same as those under normal light conditions, with no significant differences observed (*p* > 0.05). However, the pupation numbers in B2, B4, B6, RB2, and RB4 treatments showed significant differences compared to the CK group (*p* < 0.001). The pupation rates are presented in Figure 7. There were no significant differences in the pupation rates of corn borers between the R4 and R6 treatments and the CK group (*p* > 0.05). However, the pupation rates in the B2, B4, B6, RB2, and RB4 groups were significantly different from those in the CK group (*p* < 0.001).

### 3.4. Effects of Red and Blue Laser Light on Oviposition in the Corn Borer

The total egg production, daily egg production, and oviposition period of corn borers under different lighting conditions are shown in Figure 7. Due to the extremely low pupation numbers of corn borers in the B4, B6, RB2, and RB4 treatments, few pairs were formed and no eggs were laid; therefore, no corresponding data were available for these groups. In the R4 and R6 treatments, the total egg production of adult corn borers was 983 and 1040 eggs, respectively, which showed no significant difference compared to the total egg production of 966 eggs in the CK group under normal lighting conditions, with the difference remaining within 100 eggs. The daily egg production in R4 and R6 was 98 and 115 eggs, respectively, while the CK group recorded 99 eggs per day, with no significant differences observed among the three groups (*p* = 0.719), remaining within a range of 20 eggs. Regarding the oviposition period, the oviposition durations in R4 and R6 were 2.6 days and 2.8 days, respectively, which showed a significant difference compared to 3.2 days in the CK group (*p* = 0.031).

**Figure 7 insects-16-00906-f007:**
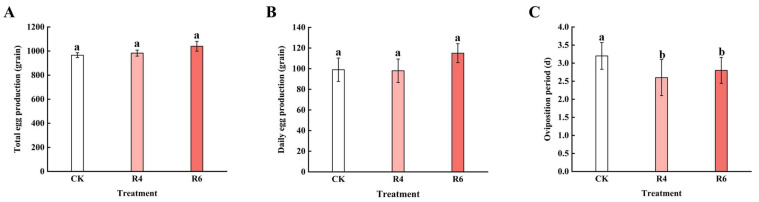
The situation of adult corn borers laying eggs: (**A**) is the total egg production. The total egg production of the CK group was 966, that of the R4 group was 983, and that of the R6 group was 1040. All groups (CK, R4, and R6) share the same letter “a”, indicating no significant difference between them. (**B**) is the daily egg production. The average daily egg production of the CK group was 99, the R4 group was 98, and the R6 group was 115. All groups (CK, R4, and R6) share the same letter “a”, indicating no significant difference. (**C**) is the egg laying cycle. The average egg laying cycle of the five adult groups in the CK group is 3.2 days, that of the R4 group is 2.6 days, and that of the R6 group is 2.8 days. Group CK is marked with “a”, while groups R4 and R6 are marked with “b”, indicating a significant difference between CK and the other two groups.

### 3.5. Effects of Red and Blue Laser Light on Hatching Rate in the Corn Borer

The hatching rates of eggs laid under different lighting conditions are shown in Figure 8. The hatching rates of eggs laid by corn borer adults in the R4 and R6 treatments were 81.6% and 79.4%, respectively, showing no significant difference when compared to the hatching rate of 71.4% in the CK group under normal lighting conditions (*p* = 0.675).

## 4. Discussion

The analysis of the effects of eight light sources on the growth and development of different life stages of the Asian corn borer revealed varying impacts of red and blue laser irradiation. At the egg stage, significant differences were observed between the laser-treated groups and the control group. Among the red light treatment groups, the R4 group showed a 1-day delay in egg hatching, with some eggs failing to hatch, while the R6 group had a 0.8-day delay. In the blue laser treatment groups, eggs in the B2 and B4 groups were delayed by 0.6 days, and those in the B5 group were delayed by 0.5 days compared to the control. In the combined red and blue laser treatment group, RB2 eggs experienced a 0.6-day delay, and RB4 eggs were delayed by 0.9 days. These findings are generally consistent with previous studies on the hatching rates of *Empoasca onukii* and fruit fly eggs, supporting the potential impact of laser treatments on pest egg development.

During the larval stage, red laser irradiation at different intensity levels did not significantly affect the pupation of the corn borer, with developmental durations comparable to those in the control group. In contrast, blue laser irradiation significantly prolonged both the larval and pupation periods. Notably, some larvae under blue light remained in the mature larval stage for up to 40 days, whereas the typical duration from egg to pupation in the CK group was approximately 20 days. Under the second intensity level of blue laser treatment, the number of pupating individuals was markedly reduced. Similarly, combined red and blue laser treatments also extended larval development and reduced pupation rates compared to both the CK and red-light-only groups. In addition to the prolonged developmental period, larvae exposed to blue light appeared more opaque, suggesting developmental stress or physiological alteration. These results indicate that blue laser light exerts a strong inhibitory effect on larval development, particularly by delaying pupation and reducing the number of individuals completing metamorphosis.

The observed developmental delay may be associated with disruptions to endocrine signaling or interference with metabolic pathways, ultimately impacting the successful transition to the pupal stage. Such effects may alter the population dynamics and field emergence patterns of the corn borer, offering a potential non-chemical strategy for pest control through delayed development and increased mortality during metamorphosis.

The experimental results indicate that red–blue laser treatment significantly reduced the pupation rate of the corn borer (from 96.67% to 5.33%). Based on the existing literature, we hypothesize that this effect may be attributed to several factors. Previous studies have shown that light-sensitive receptors such as cryptochromes and opsins play critical roles in insect phototransduction and behavioral regulation [28]. It is therefore plausible that the narrow-spectrum laser light used in this study may abnormally activate these receptors, leading to disruptions in developmental signaling pathways.

Our results suggest that laser irradiation may interfere with the endocrine regulation of metamorphosis by disrupting the biosynthesis or release of 20-hydroxyecdysone (20E), a key steroid hormone synthesized in the prothoracic gland from dietary sterols. During metamorphosis, 20E is secreted into the hemolymph through vesicle-mediated transport. The prothoracic gland integrates multiple internal and external signals to precisely control the timing and levels of 20E pulses [29]. Disruption of this regulatory network by external light stimuli could therefore suppress normal 20E signaling, potentially leading to developmental abnormalities.

High temperatures increase insect metabolic rates, leading to higher ROS production, which must be counteracted by antioxidants to prevent oxidative damage [30]. While antioxidants are upregulated during heat stress, their effectiveness in preventing oxidative damage is still unclear. Given that ROS can disrupt cellular processes, we propose that laser irradiation may similarly increase ROS levels, overwhelming antioxidant defenses and impairing development and metamorphosis. The experimental investigation of corn borer oviposition behavior under different lighting conditions revealed that red laser irradiation at intensities R4 and R6 did not significantly affect the quantity or cycle of egg-laying compared to normal light conditions, indicating a relatively weak influence of red light on oviposition behavior. Additionally, while red light irradiation did not significantly alter hatching rates, it appeared to support normal reproductive behavior to some extent.

In this study, corn borers were exposed to red laser light at varying intensities (R4 and R6) to assess the impact on oviposition and egg hatching performance. No significant differences were observed in the number of eggs laid or the hatching rates between the treatment groups and the normal light control group. This suggests that red light, within the intensity range used, has a limited effect on the reproductive behavior of corn borers. This lack of impact may be attributed to the insect’s visual system, which is primarily sensitive to ultraviolet and blue light, with minimal response to long-wavelength red light. Furthermore, red light may not significantly influence the secretion of endocrine hormones, such as juvenile hormones and ecdysteroids, that regulate reproduction.

While the laboratory results demonstrate the biological efficacy of laser treatments in inhibiting larval development and reproductive capacity, their practical application as a physical pest control strategy in field conditions requires further investigation. Notably, *O. furnacalis* larvae typically bore into corn stalks and ears, raising concerns about the ability of laser beams to penetrate plant tissues and effectively reach the concealed targets. Although plant epidermal layers may attenuate or scatter laser energy, existing studies indicate that certain wavelengths, particularly in the red and near-infrared (NIR) ranges, can penetrate plant tissues to a certain depth [31,32]. For example, laser-induced breakdown spectroscopy (LIBS) has been successfully applied to analyze the internal composition of plant materials, demonstrating that laser pulses are capable of interacting with deeper tissue layers.

Moreover, the scalability of laser-based interventions in open-field agriculture remains uncertain. Challenges such as beam targeting under variable environmental conditions, energy consumption, automation requirements, and safety constraints must be addressed. These limitations suggest that, while laser irradiation holds promise, particularly in controlled environments such as greenhouses or storage facilities, its widespread field deployment may necessitate the development of integrated delivery systems, such as autonomous robotic platforms or drone-mounted lasers, capable of overcoming these challenges.

Future research should explore the engineering feasibility of such delivery systems, assess potential non-target effects, and evaluate long-term ecological and economic outcomes of laser-based pest management strategies. These findings offer valuable insights into how light environments influence the behavior of agricultural pests and provide important reference material for further research. Moreover, they contribute theoretical support for the development of light-based pest control technologies. While both red–blue lasers and LEDs can induce diapause in corn borer larvae, laser treatment demonstrated significantly lower energy consumption per unit treatment area compared to LED irradiation. Additionally, the extended operational lifespan of laser systems suggests that their large-scale field application could result in substantial cost reductions.

## 5. Conclusions

This study presents preliminary evidence that blue laser light can significantly inhibit the development and reproduction of the corn borer, particularly in terms of egg hatching, larval growth, pupation, and oviposition. These findings support the potential application of laser light as a non-chemical pest management strategy. The laser system used in this research, originally designed as a supplementary plant growth light, was found to not only enhance crop yields in greenhouse and open-field conditions but also to reduce pest infestations. Specifically, it decreased mite populations on hardy kiwifruit (*Actinidia arguta* (Siebold and Zucc.) Planch. ex Miq.) leaves and suppressed gray mold in strawberries.

Although the laser system has yet to be tested in maize field conditions, these promising results suggest its potential for integration into integrated pest management (IPM) programs in maize production systems. Future work will focus on optimizing the laser parameters for use in maize greenhouses and field trials, with the aim of improving both crop yield and pest control efficacy. While further engineering and field validation are needed, our findings lay a foundation for the potential integration of laser-based methods into sustainable pest management systems.

## Figures and Tables

**Figure 1 insects-16-00906-f001:**
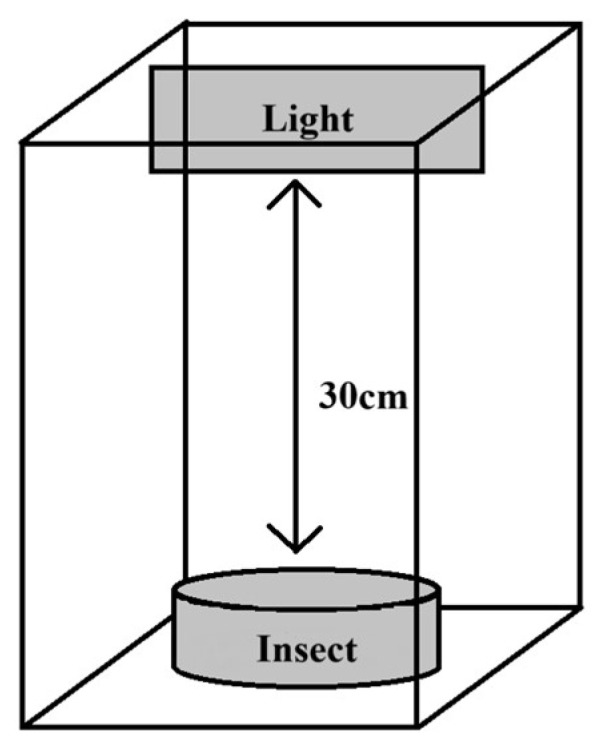
Schematic diagram of the experimental setup. The size of the dark box is 43 cm × 33 cm × 38 cm, wrapped in opaque black cloth on the outer layer, and the laser light is 30 cm away from the insect source.

**Figure 2 insects-16-00906-f002:**
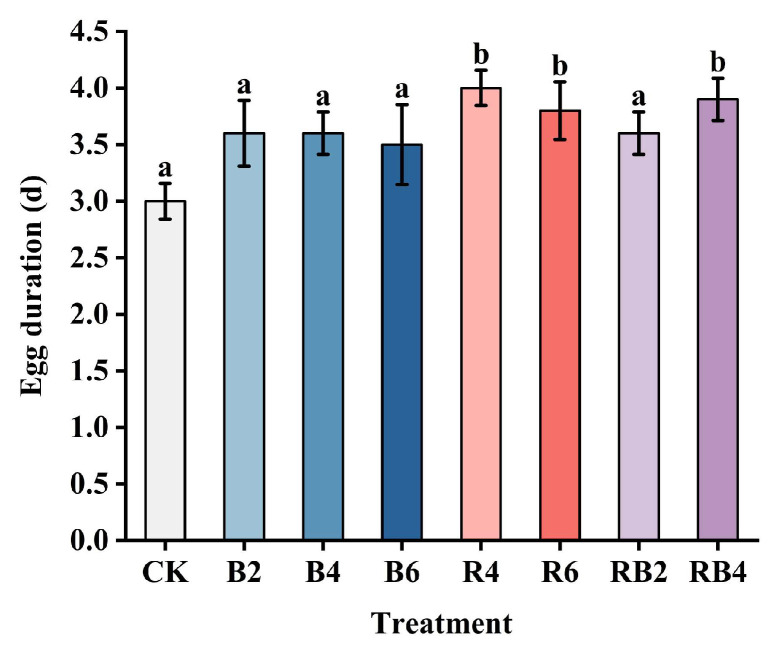
Hatching times of insect eggs under different lighting conditions. Under red light treatment, the hatching of R4 and R6 eggs was delayed by 1 day and 0.8 days, respectively, compared to the CK group. In blue light treatments, the hatching times of B2 and B4 eggs were delayed by 0.6 days compared to CK, and B6 eggs were delayed by 0.5 days. In the red and blue laser coupling treatment group, the eggs of RB2 and RB4 were delayed by 0.6 days and 0.9 days, respectively, compared to the CK group. Different letters indicate significant differences between groups (Tukey’s HSD, *p* < 0.05).

**Figure 3 insects-16-00906-f003:**
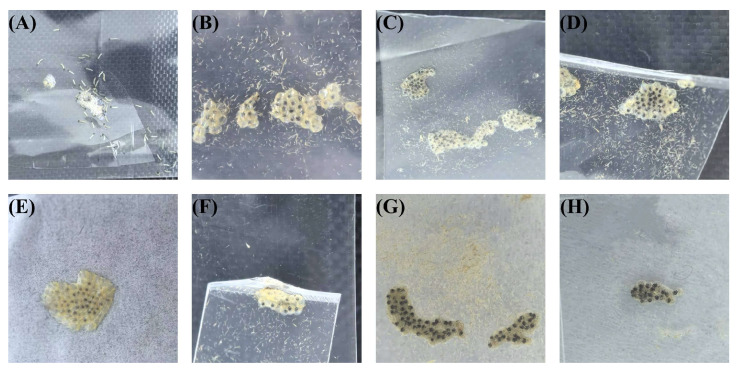
The status of insect eggs in different treatment groups when larvae hatched in the CK group, where (**A**) is CK; (**B**) is B2; (**C**) is B4; (**D**) is B6; (**E**) is R4; (**F**) is R6; (**G**) is RB2; and (**H**) is RB4.

**Figure 4 insects-16-00906-f004:**
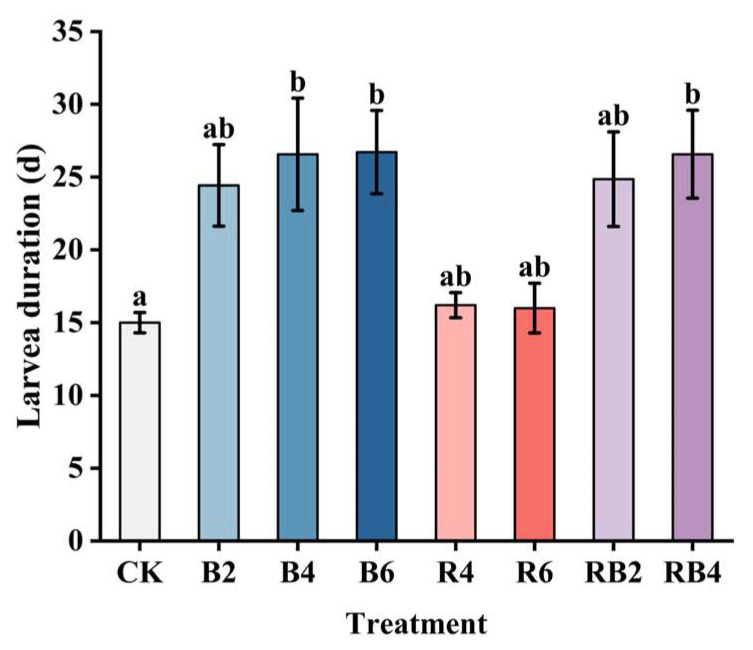
Larval stages of corn borer under different light source conditions. The larval stage in the figure is the average value of each group. The treatment group containing blue light significantly prolonged the larval stage, with B2, B4, and B6 all extending for more than ten days and RB2 and RB4 also extending for about ten days. Different letters indicate significant differences between groups (Tukey’s HSD, *p* < 0.05).

**Figure 5 insects-16-00906-f005:**
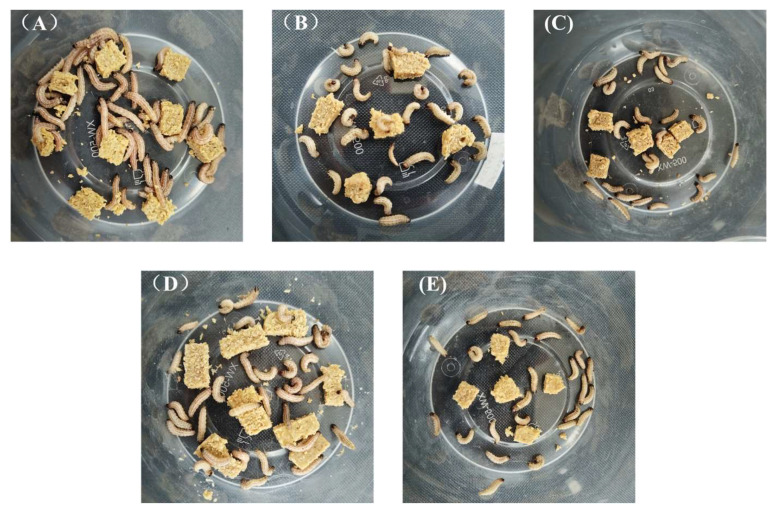
Corn borer larvae containing blue light treatments with a larval stage longer than 30 days, among them (**A**) is B2; (**B**) is B4; (**C**) is B6; (**D**) is RB2; and (**E**) is RB4.

**Figure 6 insects-16-00906-f006:**
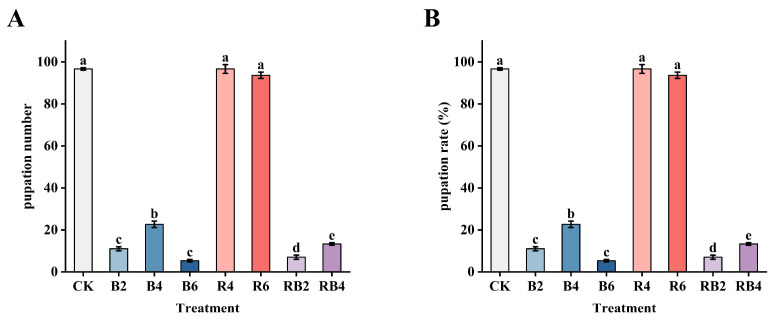
(**A**) The number of pupae: the number of pupae in CK group was 97, and the number of pupae in the blue light treatment group (B2, B4, B6) was significantly reduced, which were 11, 23, and 5, respectively. The red light and red blue light treatment groups (R4, R6) were 97 and 94. The red and blue light treatment groups (RB2, RB4) were 7 and 13. (**B**) The rate of pupation: the number of pupae in the CK group was 96.67%, and the pupation rate in the blue light treatment group (B2, B4, B6) were significantly reduced, which were 11%, 22.67%, and 5.33%, respectively. The red light and red blue light treatment groups (R4, R6) were 96.67% and 93.67%. The red and blue light treatment groups (RB2, RB4) were 7% and 13.33%. Different letters indicate significant differences between groups (Tukey’s HSD, *p* < 0.05).

**Figure 8 insects-16-00906-f008:**
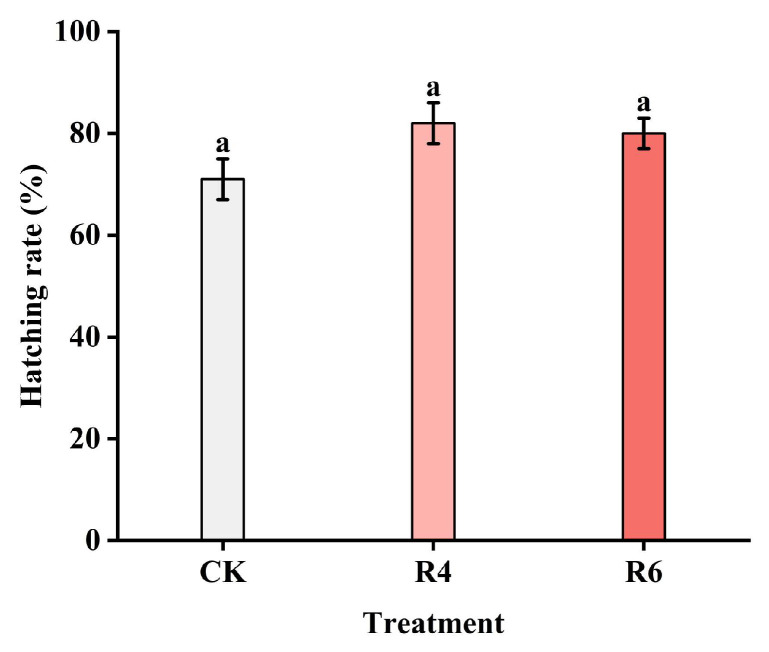
Hatching rate of corn borer eggs under different light conditions. The hatching rate of the CK group was 71.4%, that of the R4 group was 81.6%, and that of the R6 group was 79.4%. All groups (CK, R4, and R6) share the same letter “a”, indicating no significant difference between them.

**Table 1 insects-16-00906-t001:** Artificial diet formula for the Asian corn borer.

Ingredients	Companies/Manufacturers	Quantity
Wheat germ	Jiahui Feed, Shijiazhuang, China	450 g
Yeast powder	Xi’an Yibo Trading Co., Ltd., Xi’an, China	150 g
Sucrose	Tianjin Huasheng Chemical Reagent Co., Ltd., Tianjin, China	45 g
Agar powder	Taizhou Siqi Teaching Instrument Co., Ltd., Taizhou, China	48 g
Sorbic acid (C_6_H_8_O_2_)	Xuanhao Wen Food Shop, Fuping County, Baoding, China	12 g
Methylparaben (C_8_H_8_O_3_)	Henan Gaobao Industrial Co., Ltd., Zhengzhou, China	12 g
Ascorbic acid (C_6_H_8_O_6_)	Hefei Qianfang Animal Health Technology Co., Ltd., Hefei, China	12 g
Distilled water	Jinan Lixia Changlei Chemical Business Unit, Jinan, China	1600 ± 100 mL

**Table 2 insects-16-00906-t002:** Comparative analysis of laser versus LED light sources. (The comparative data in this table are derived from greenhouse applications).

Light Source	Laser Plant Filler Light	LED
	(Greenhouse Application)	(Greenhouse Application)
Watt per acre (kW)	0.2 (10 Units)	30 kW (75 Units)
Annual energy use per unit area (kWh/acre/year)	564	43,800
Power per individual lamp (W)	13–20	400
Photosynthetic photon flux (μmol/s)	10–15	622–1655
PPFD (μmol/s/m^2^)	0.2–0.6	Calculation based on crop requirements
Annual energy use per unit area (kWh/acre/year)	0.2 (10 Units)	30 kW (75 Units)
Power per individual lamp (W)	564	43800
Photosynthetic photon flux (μmol/s)	13–20	400

**Table 3 insects-16-00906-t003:** Light sources and parameters (PAR stands for photosynthetically active radiation, with units of μW/cm^2^. PPFD is the photosynthetic photon flux density, measured in μmol/m^2^/s).

Group	PAR	PPFD	PPFD Red	PPFD Blue	PPFDUVB	PPFD Green
CK	274.776	10.877	3.275	3.196	0.211	4.406
R6	453.171	25.138	25.138	0	0	0
R4	249.381	13.851	13.851	0	0	0
RB4	500.952	24.116	16.813	7.303	0	0
RB2	392.796	17.129	9.489	7.640	0	0
B6	435.151	16.230	0	16.230	0	0
B4	318.814	12.114	0	12.114	0	0
B2	99.539	3.686	0	3.686	0	0

Note: treatment codes (e.g., R4, B2) indicate relative preset intensity levels of red or blue laser light, corresponding to the absolute irradiance values listed in the table.

## Data Availability

The data presented in this study are fully contained within the figures of the article. Further inquiries can be directed to the corresponding author.

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
