# Peer review of "Effects of Red and Blue Laser Irradiation on the Growth and Development of Ostrinia furnacalis"

_insects, 2025, doi:10.3390/insects16090906_

Round 1

Reviewer 1 Report

Comments and Suggestions for Authors

Liang et al. investigated the influence of red and blue laser irradiation on the growth and development of the Asian corn borer (Ostrinia furnacalis (Guenée)). Their findings indicate that blue laser light significantly prolongs larval development and reduces pupation rates. The study highlights an innovative and potentially promising approach to physical pest control. The topic is timely and of interest to Insects readers, especially given the growing demand for sustainable alternatives to chemical pesticides. The experimental design is generally sound, and the results are supported by statistical analysis. However, substantial revisions are necessary before the manuscript can be considered for publication.

Major Comments:

1. The authors reared larvae under lights of different intensities and spectra, finding that blue laser light markedly extended larval development time and reduced pupation rates. While they interpret these results as evidence of a promising eco-friendly alternative to chemical insecticides, the practical feasibility of this method as a physical control strategy is questionable. O. furnacalis larvae typically bore into corn ears or stalks, raising concerns about whether laser light can penetrate plant epidermis. Furthermore, the scalability of this approach in open-field conditions remains untested and should be critically discussed.

2. Although the use of laser light for pest suppression is novel, the Introduction should better differentiate this work from previous studies using LEDs or broad-spectrum light sources. At present, the literature review combines laser-based and non-laser optical research, which risks overstating the novelty of the approach.

3. The description of the “light intensity levels” (R4, R6, B2, etc.) is unclear to readers unfamiliar with the setup. The manuscript should clearly define what these intensity levels represent in absolute terms and explain the rationale for their selection.

4. The Discussion proposes several speculative mechanisms (e.g., cryptochrome/opsin activation, inhibition of 20-hydroxyecdysone synthesis, ROS accumulation) without direct experimental evidence from this study. These hypotheses should be explicitly framed as speculative and supported by appropriate citations.

Minor comments:

Table 1   In the artificial diet formula, specify the unit of measurement for each ingredient.

Line 199 p should be italicized.

Figure 2, 7 and 8  Statistical significance is not indicated; use letters or asterisks to denote differences.

Figure 4 and 6     The current method for indicating differences is non-standard. Asterisks should be used only for pairwise comparisons, not above each individual bar. Using different letters to denote significant differences is recommended.

Figure 5 The image lacks a scale bar.

Line 246 Some p-values are reported as “p = 1” for non-significant differences. While mathematically possible, this is uncommon and may reflect a rounding or reporting issue.

Line 426 Chinese text should not appear in the references.

There are numerous formatting irregularities in the references; please revise each one according to the journal’s requirements.

Reviewer 2 Report

Comments and Suggestions for Authors

This is an interesting and novel paper that explores the impact of different wavelengths of light from laser lighting on developmental traits of the oriental corn borer. The methods are sound and the results are well supported by the data presented. I have only a few points of correction or questions as follows:

Materials and Methods: you mention that rearing was conducted under a 12 light 12 dark photoperiod. Is this the same pattern for the treatments or was the laser light continuous?

line 94 Drosophila melanogaster needs italics

Table 3. The individual blue B4 treatment shows a PPFD of 12.114 but in the combined RB4 treatment the blue PPFD is only 7.303 -- shouldn't these levels be the same?

Round 2

Reviewer 1 Report

Comments and Suggestions for Authors

I thank the author for their diligent efforts in revising the manuscript. All of my concerns have been fully addressed, and the overall quality of the paper has improved substantially. I therefore recommend acceptance of the manuscript.